# Protocol for the PLAY Study: a randomised controlled trial of an intervention to improve infant development by encouraging maternal self-efficacy using behavioural feedback

Alessandra Prioreschi ![ORCID],[1] Rebecca Pearson,[1,2] Linda Richter,[3] Fiona Bennin,[1] Helene Theunissen,[1] Sarah J Cantrell,[1] Dumsile Maduna,[1] Deborah Lawlor,[4] Shane A Norris ![ORCID] [1]

For numbered affiliations see end of article.

**Correspondence to**
Dr Alessandra Prioreschi;
alessandra.prioreschi@wits.ac.za

## ABSTRACT

**Introduction** The early infant caregiving environment is crucial in the formation of parent–child relationships, neurobehavioural development and thus child outcomes. This protocol describes the Play Love And You (PLAY) Study, a phase 1 trial of an intervention designed to promote infant development through encouraging maternal self-efficacy using behavioural feedback, and supportive interventions.

**Methods and analysis** 210 mother–infant pairs will be recruited at delivery from community clinics in Soweto, South Africa, and individually randomised (1:1) into two groups. The trial will consist of a standard of care arm and an intervention arm. The intervention will start at birth and end at 12 months, and outcome assessments will be made when the infants are 0, 6 and 12 months of age. The intervention will be delivered by community health helpers using an app with resource material, telephone calls, in person visits and behavioural feedback with individualised support. Every 4 months, mothers in the intervention group will receive rapid feedback via the app and in person on their infant's movement behaviours and on their interaction styles with their infant. At recruitment, and again at 4 months, mothers will be screened for mental health risk and women who score in the high-risk category will receive an individual counselling session from a licensed psychologist, followed by referral and continued support as necessary. The primary outcome is efficacy of the intervention in improving maternal self-efficacy, and the secondary outcomes are infant development at 12 months, and feasibility and acceptability of each component of the intervention.

**Ethics and dissemination** The PLAY Study has received ethical approval from the Human Research Ethics Committee of the University of the Witwatersrand (M220217). Participants will be provided with an information sheet and required to provide written consent prior to being enrolled. Study results will be shared via publication in peer-reviewed journals, conference presentation and media engagement.

**Trial registration number** This trial was registered with the Pan African Clinical Trials Registry (https://

## STRENGTHS AND LIMITATIONS OF THIS STUDY

⇒ The Play Love And You Study intervention uses behavioural feedback linked with personalised guidance to improve and individualise intervention content and implementation.

⇒ The study will be guided by an advisory group of community members, consisting of women with infants and their mothers/influential family matriarchs, and consultation with this advisory group will happen before and during the intervention to ensure we remain contextually aware in creating and delivering our study content.

⇒ The study uses advanced and robust measurement tools, which in some cases are novel in South Africa and/or globally, and thus should be able to generate reliable evidence that can inform future research.

⇒ The study relies to some extent on the availability of cell phone or computer access for the delivery of intervention content, which may be limiting in the Soweto setting.

⇒ The COVID-19 pandemic may interfere with our recruitment and intervention strategies, and we have made the study less reliant on face-to-face interactions to accommodate any further lockdown restrictions in South Africa.

pactr.samrc.ac.za) on 10 February 2022 (identifier: PACTR202202747620052).

## INTRODUCTION

Over the last decade, emphasis globally is being placed on improving early childhood development (ECD).[1] Estimates are that 250 million children (43%) under the age of 5 living in low-and-middle-income countries (LMICs) are at risk of not meeting their developmental potential.[2] Furthermore, a double burden of malnutrition persists in LMICs, with wasting present in 6.4% and

overweight present in 6% of children under the age of 5.[3] While optimising growth and development requires a life course approach, the first 2 years are key in optimising developmental potential.[1 4] This highly sensitive period (the first 1000 days of life) is an experience-dependent and experience-expectant process, which is plastic and malleable.[5] The WHO has recently published four key guidelines for improving ECD[6]: integration of nutrition and caregiving interventions, promotion of early learning, supporting maternal mental health and responsive caregiving.

### Integrating nutrition into caregiving interventions

In the first 6 months of an infant's life, exclusive breast feeding is regarded as the optimal and most nutritionally complete form of infant nutrition.[7 8] Research indicates that infants who are breast fed for longer may have higher cognitive function, and lower infectious morbidity and mortality. Breast feeding may also protect against diabetes, cardiometabolic disease and overweight later in life.[7] It is estimated that improving global breastfeeding rates alone could potentially prevent more than 800 000 children deaths annually.[7] Despite the widely established benefits of breast feeding, in the South African context, roughly 67% of infants initiated breast feeding within the first hour of birth in 2016, yet only 32% of children under the age of 6 months remain exclusively breast fed. The mean duration of exclusive breast feeding was found to be only 2.9 months,[9] with widespread introduction of solid foods and non-breastmilk liquids before 3 months.[10 11] Breastfeeding self-efficacy has been found to directly influence breastfeeding practices, and high breastfeeding self-efficacy was found to play a critical role in exclusive breastfeeding duration in the South African setting.[12]

Qualitative findings from urban South Africa reported that infant nutrition interventions in this context should focus on: fostering maternal self-efficacy around exclusive breast-feeding; challenging mixed feeding practices and encouraging more responsive feeding approaches; and engaging family members to promote supportive household and community structures around infant feeding.[13]

### Promotion of early learning through play

Infant movement behaviours have been identified as important for early childhood growth and development.[14] Three categories of movement behaviours are of key importance: physical activity, sleep and sedentary behaviours.[14] These three behaviours are contingent on each other to make up a 24-hour day, yet have independent health implications and thus independent recommendations, which should all be met for optimal health. Guidelines for these behaviours in the first few years of life encourage tummy time and active play, sufficient and regular sleep, and minimisation of activities that require infants to be restrained (eg, car seats, strollers, walkers).[14–16] Play is defined as intrinsically motivated activities done for recreational purposes and enjoyment.

Play is how children learn, explore, engage in physical activity and motor development, develop socially, emotionally and cognitively, and bond with caregivers.[17 18] Outdoor play is particularly encouraged due to its beneficial effects on both physical activity and development.[19] Additionally, interactive and stimulating play (such as reading, singing and playing with other adults or children) is encouraged to promote development[20] and also supports early learning.[6]

Formative research in South Africa has determined that higher intensity movement was associated with earlier attainment of developmental milestones[21] and with taller stature and lower abdominal adiposity.[22] However, the majority of infants in South Africa were not meeting recommended 24-hour movement guidelines.[23] When using synchronous 24-hour accelerometery worn by mothers and infants, we showed that increased physical activity among mothers while caring for their infants was associated with increased infant movement during the same period of time.[24] This highlights the important role of the mother in infant movement behaviours, and thus growth and development. Interactive play is particularly important for promotion of parent–child bonding, offering an ideal opportunity for parents to engage with their children through stimulating activities which encourage movement, and in turn promote cognitive, emotional and social development. Stimulation interventions in LMICs have been shown to improve home caregiving environments, mother–child interactions and maternal health literacy around ECD.[25]

Qualitative research examining maternal perceptions of the importance of play in infancy reported that mothers in South Africa perceive developmental attainment as a key child outcome, indicating that interventions in this context should focus on promoting child development.[26] Mothers also reported significant barriers to encouraging play, including financial constraints, safety concerns and lack of certainty about how to promote play. Importantly, they did not perceive infant movement behaviours as behaviours that could be, or needed to be changed.[26] Additionally, and similar to findings from national reports, mothers reported not reading to their children, and most children in this context did not attend creche, indicating the importance of intervening in the home environment to improve opportunities for early learning.[26 27]

### Supporting maternal mental health

In South Africa, up to 57% of mothers present with symptoms of postnatal depression.[28] Qualitative studies have shown that in Soweto, household centred issues impacted on the amount of chronic anger, stress, depression and suicidal ideation as well as material and relational hardships experienced by women.[29] Relational stress is exacerbated by the fact that mothers and children in South Africa live in multiple-generational households (66%), while only 21% are said to live in households that may be defined as nuclear.[27] Women in Soweto are also exposed

to high unemployment rates, low salaries, rife domestic violence[28] and thus social vulnerability.[30]

Infants of mothers with depressive symptoms have an increased chance of developmental delay in the first 12 months of life.[31] In South Africa, a significant increase in psychological challenges at 2 years of age was evident when maternal postpartum depression was present.[31] Additionally, within the context of high exposure to trauma in South Africa, post-traumatic stress disorder in mothers was significantly associated with poorer infant developmental outcomes in fine motor and adaptive behaviour, as well as emotional regulation.[32 33] Due to the high prevalence of poor postpartum mental health in South Africa, and lack of support available for these women, screening of mothers for postpartum depression, anxiety and post-traumatic stress disorder is recommended.[28]

### Responsive caregiving

Responsive caregiving is essential in helping children to thrive. Responsive caregiving refers to the encouragement of caregivers to make eye contact, to smile, cuddle and praise infants. Responsive parenting urges caregivers to notice their children's cues of hunger, illness, emotional distress and signs of pleasure and to respond sensitively and appropriately,[20] while also allowing the child to independently explore and self-regulate. Responsive caregiving is therefore important for ensuring better nutrition and feeding practices, health practices, and development. Responsive caregiving ultimately leads to improved bonding between mother and infant. In a well-bonded relationship, the mother aims to meet her infant's needs and the infant responds to having its needs met. Bonding serves to generate the feeling of connectedness between the dyad,[34] and is crucial for developing a secure infant attachment.

A recent systematic review and meta-analysis of parenting interventions to improve ECD in children under the age of 5 has shown that interventions with responsive caregiving components had greater effects on child cognitive development, as well as on parenting practices and parent–child interactions.[35] The authors recommended that future interventions should aim to directly support parent responsiveness and sensitivity.

### Tools for supporting parents in promoting these behaviours

The early infant caregiving environment is thus crucial in the formation of parent–child relationships, mother–infant synchronicity and contingency, behaviour development and therefore child outcomes.[20 36 37] Mothers who are supported in their transition to parenthood have higher self-efficacy and are less likely to develop postpartum depression.[35 38] Infants who have confident and healthy mothers are more likely to develop resilience and emotional and cognitive competencies, to have healthy growth trajectories and to develop healthy behavioural patterns.[39]

Maternal self-efficacy is a likely contributor to responsiveness and sensitivity and is determined by a mother's belief in her ability to care for her child, her emotional and psychological health, her child's temperament and health and various environmental factors. Strategies to improve self-efficacy in parenthood include improving health literacy, reminding mothers of their natural ability to care for their child, social support and promotion of mental health. Interventions that are targeted and focus specifically on changing maternal behaviour are most effective.[40] Maternal mental health should be supported through integration into ECD interventions and programmes, yet a lack of studies on maternal mental health that report on child health and development outcomes, in addition to mental health outcomes, has been identified as a critical gap.[6] Other gaps include identifying the most effective responsive caregiving interventions in LMICs, combining nutrition and caregiving strategies[6] and identifying ways to improve parenting behaviours without undermining self-efficacy.

In the first 3-month postpartum, tools to sensitise mothers to infant's competencies and to promote positive and nurturing interactions between caregivers and infants are successful in reducing postpartum depression, and increasing self-efficacy in both parents and healthcare providers.[1 6 41] Following early supportive intervention and sensitisation to infant behavioural cues, mothers benefit from continued support around breast feeding, and promotion of healthy growth and development.[1 6] This can include provision of content to improve health literacy, practical tools and activities that promote infant development, continued sensitisation to infant behaviours and mother–infant interaction styles and mental health promotion.

Behavioural feedback is an important and effective component of parenting interventions and behaviour change interventions in general,[42] which can help to enhance parenting skills and thus child outcomes.[41 43] Feedback consists of comparing individual behaviour to a standard and is thought to change behaviour by changing the individual's locus of control and focusing attention on a particular behaviour, thus regulating that behaviour.[44–46] Feedback on parenting behaviours can be directive and/or responsive to behaviours in real-time, or following review of video recorded interactions[43 47]; and encouraging parents to practice skills following feedback is one of the strongest and most consistent predictors of improved parenting outcomes in meta-analyses.[40 41] Additionally, tailoring behavioural feedback is particularly effective for behaviour change in general.[48]

### Aims

This protocol outlines a phase 1 trial with the following aims:

1. To test the efficacy, feasibility and acceptability of an intervention designed to promote development through encouraging maternal self-efficacy using behavioural feedback, content provision and mental health support.

2. To test the efficacy, feasibility and acceptability of providing content, support and referrals to promote breastfeeding self-efficacy.
3. To test the efficacy, feasibility and acceptability of providing content, and increasing opportunities for early learning through play to promote health literacy.

## Hypotheses

1. Using behavioural feedback and content provision to support mothers in their ability to be responsive, promote early learning in the home, understand and engage with their infants' health behaviours (specifically movement behaviours interactions), in conjunction with supporting mothers' mental health, will result in improved maternal self-efficacy, and therefore better infant development in the first year of life.
2. Supporting mothers with breastfeeding content, guidance and referrals to lactation specialists in the first 6-month postpartum will increase self-efficacy, and thus improve breastfeeding exclusivity.
3. Proving mothers with access to resources, content and activities to support their infant's play and development will increase their health literacy, and thus encourage play and promote development.

## METHODS AND ANALYSIS

The Play Love And You (PLAY) Study was developed with the overriding theme that all an infant really needs are opportunities for movement and development (play), responsive and interactive caregiving (love) and the mother herself (you).

### Study design

The PLAY Study was designed following the Standard Protocol Items: Recommendations for Interventional Trials 2013 statement.[49] Mother–infant pairs (n=210) will be recruited starting in the third quarter of 2022 until the second quarter of 2023, and individually randomised into two groups (total n after expected attrition=150, n=75 in each group) using simple 1:1 randomisation generated by Stata V.17. The intervention will last from birth to 12 months, with main assessments made when the infants are 0, 6 and 12 months of age. The trial will consist of a standard of care arm and an intervention arm—see figure 1. Data collection teams will be blinded to randomisation at baseline and during primary outcome assessments, and separate data collection teams will collect baseline compared with primary outcome data. Blinding will be maintained by the data manager and broken only for a priori defined analyses (figure 2). Randomisation

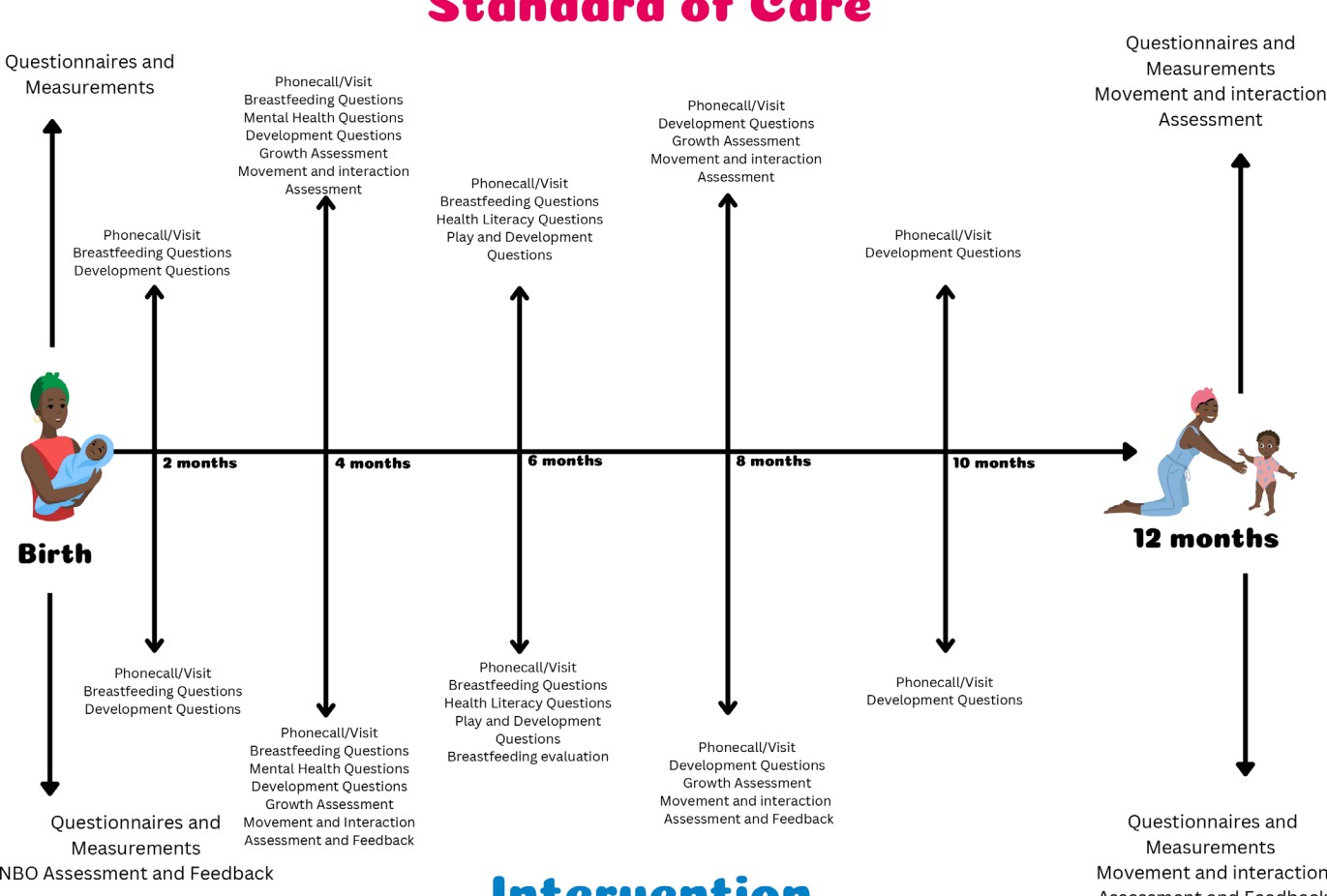

**Figure 1** Participant flow chart. NBO, Newborn Behavioural Assessment.

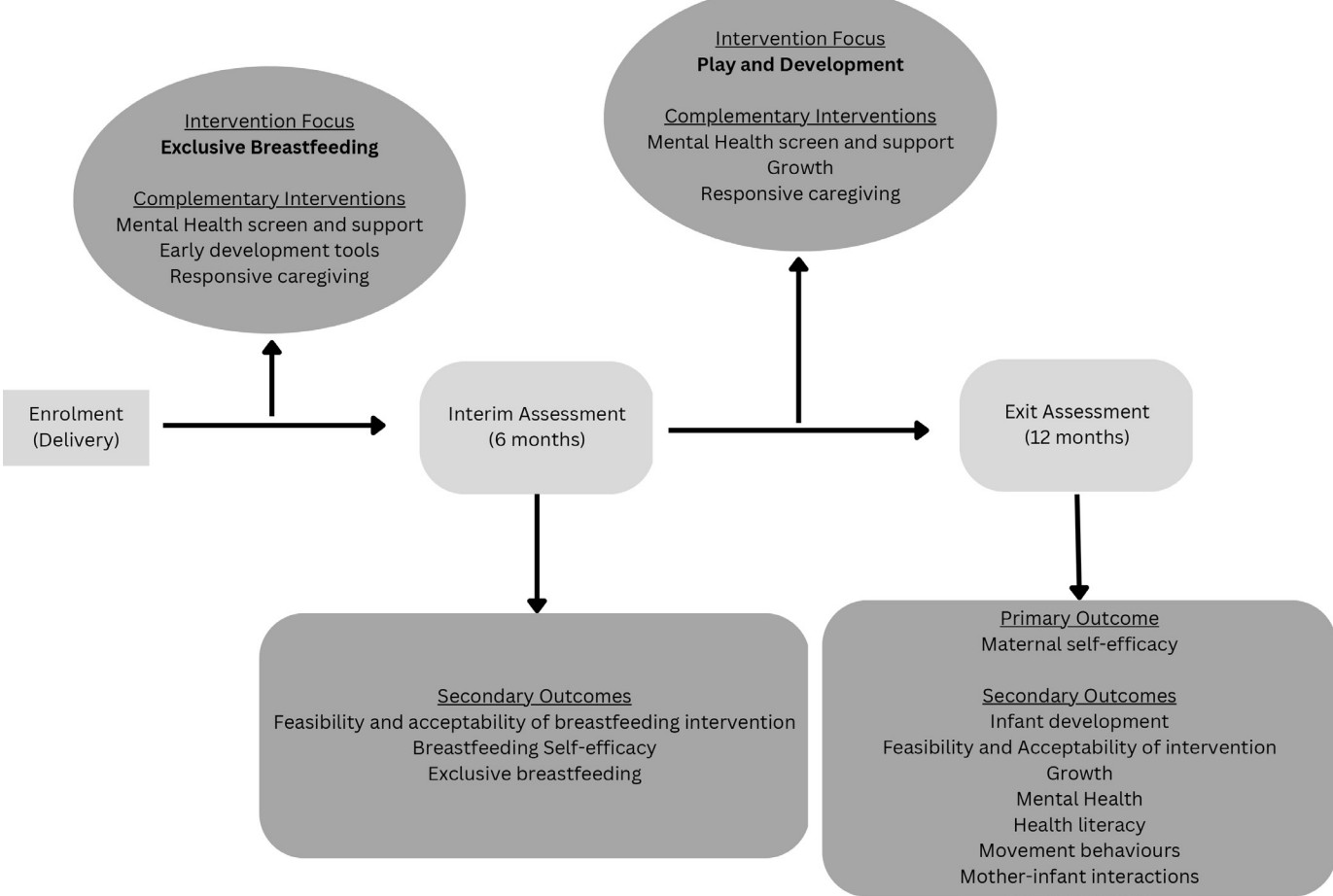

**Figure 2** Intervention themes and assessments.

into either group will be generated by the data manager only after recruitment and baseline assessments are completed, whereafter participants will be assigned to the intervention delivery teams accordingly (intervention delivery teams will thus not be blinded). Intervention delivery teams will work separately and independently of each other, and of data collection teams. The trial is due to be completed in the second quarter of 2024.

## Participants

We will recruit 210 mother–infant pairs within 3–10 days following delivery at community clinics in Soweto. Soweto is an urban-poor area in the city of Johannesburg covering 200 km$^2$ with over 1.3 million people (6400/km$^2$). Mother–infant pairs will be eligible to participate if:
► Mothers agree to participate for the duration of the study.
► They reside in Soweto and plan to remain in Soweto for the study duration.
► The mother is ≥18 years of age.
► The mother is the primary caregiver.

## Trial interventions

### Standard of care arm

Mothers assigned to the standard of care arm will receive the Road to Health booklet as well as the Side-by-Side

services provided by the National Department of Health as routine postpartum care. This content is divided into five knowledge pillars (nutrition, love, protection, healthcare and extra care) and functions to record child's growth and health interventions, provide information for caregivers and encourage collaboration between healthcare workers and caregivers (https://sidebyside.co.za).

In addition to the standard of care, they will be contacted telephonically or in person every 2 months to check in, and to provide referrals to healthcare facilities and community services if necessary; and will have anthropometric, development, movement and interaction measurements taken every 4 months.

The standard of care arm will therefore include:
► Road to Health booklet as well as the Side-by-Side services provided as routine care postpartum at delivery.
► Telephonic or in person check in every 2 months, and referrals to healthcare facilities and community services if necessary.
► Measurements taken every 4 months.

### Intervention arm

Mothers assigned to the intervention arm will receive everything as per the standard of care arm. Additionally,

they will receive various microinterventions according to the following key intervention themes: (1) exclusive breast feeding for 6 months, (2) interactive play and promotion of early learning, (3) responsive caregiving and (4) maternal mental health. Each intervention theme will be prioritised during different phases of the trial and supported by components from the other themes. Figure 2 details when each theme will be prioritised and which supporting interventions will be included. Interventions will be delivered telephonically or in person, and via a mobile app created for the purposes of this trial. Interventions are detailed below.

### Behavioural feedback

At recruitment, mothers will have the Newborn Behavioural Assessment (NBO)[50] administered by trained research staff. The NBO consists of 18 neurobehavioural observations along four dimensions—autonomic, motor, state organisation and attentional–interactional—ultimately yielding a profile of the infant's behavioural repertoire. Importantly, the NBO allows for provision of infant specific anticipatory guidance, thus engaging parents in discussions around nurturing care, mother–infant interaction, play and development, and breast feeding. The NBO has been shown to improve maternal sensitivity, and to promote positive maternal–infant interactions.[50]

Every 4 months, mothers will receive rapid feedback via the PLAY Study app and in person on their infant's and their own movement behaviours and on their interaction styles with their infant. Feedback will be based on objective measurement of infant movement behaviours and headcam video footage of interactive play in the home environment. Feedback on movement behaviours will consist of a graphical representation of the infant's and the mother's movement behaviours over the previous week, with personalised information and advice about: sleep routines, play and tummy time, nap times, daily activity routines, peaks of high activity versus lulls in activity, and recommended daily movement guidelines. This feedback is designed to sensitise mothers to their infant's behaviour patterns and to encourage healthy infant movement and routines, thus improving their self-efficacy to promote healthy routines for development. An example of the graphical feedback presented is included in online supplemental figure 1. Feedback on mother–infant interactions during play is guided by the Video-feedback intervention to promote positive parenting and sensitive discipline (VIPP-SD)[51] and will consist of a video segment from a peak interaction moment with mother and infant perspectives presented side-by-side, with personalised information and guidance on: positive interactions at 4 months (baby smiling, eye contact, soothing, directing attention, reinforcing mothers strengths, communication and bonding), perspective taking at 8 months (describing and narrating the interaction, seeing baby as an individual, understanding baby cues) and recognising child feedback at 12 months (sensitivity and responsiveness, following cues, adjusting behaviour,

soothing styles, interaction chains). This feedback is designed to promote positive and nurturing interactions and improve maternal self-efficacy. The app will contain the graphical and video feedback and will also allow mothers to observe changes in their infant's behaviour and development over the trial period.

### Mental health screening and support

At recruitment, and at 4 months mothers will complete the Edinburgh Postnatal Depression Screening (EPDS) Questionnair, the Post-Traumatic Stress Syndrome Checklist 5 (PCL-5), the Adverse Childhood Experiences (ACE) Scale and the Antenatal Stress Questionnaire (ASQ). Women who score in the high-risk category on one or more of these questionnaires will receive a phone call to assess any immediate risk, and referral to an individual counselling session from a licensed psychologist at their local clinic, followed by continued support as necessary. Women who score at medium risk on one or more of these questionnaires will receive referrals to the licensed psychologist and continued support as necessary, and in both cases uptake of this support will be monitored. All participants will receive a flyer on mental health and available services and support upon completion of the questionnaires.

### Health literacy content delivered via the PLAY Study mobile app

Every week, health theme-specific literacy content will be delivered via the PLAY Study mobile app, which was designed for the purposes of this intervention. The app creates a password-protected profile for each participant and houses a section to deliver the feedback when available, as well as the weekly (and historic) content. This content is loaded to the profile every Monday, and participants can choose to receive push notifications to alert them of new content. Participants can engage with the app at any time, for as long as they wish and as many times as they wish during the intervention. The weekly content will appear in the format of quick and easy tips, short videos, infographics or ideas for home-based activities. Mothers will have continuous access to the app with all relevant resources, activities and tools to support their interactions with their infant.

### Health literacy content delivered telephonically or in person

Every 2 months, theme-focused content will be delivered telephonically or in person. Telephonic or in person visits will also follow the key intervention themes schedule, but will allow mothers to guide the conversation by using open questions and providing anticipatory guidance. Referrals to lactation specialists or psychologists or other services will be provided as necessary.

Health literacy content themes will be scheduled as follows and according to figure 2.

► From 0 to 6 months, content will be designed to increase breastfeeding self-efficacy and therefore promote exclusive breast feeding.

- ► From 6 to 12 months, content will be designed to increase health literacy about play and development by providing content and activities to support and therefore improve these outcomes.
- ► From 0 to 12 months, there will be additional content to support maternal mental health, responsive caregiving and bonding to increase self-efficacy and therefore improve infant development.

All content will be guided by a broad range of pre-existing content such as UNICEF resources (https://www.unicef.org) and WHO recommendations[6] and other evidence based online resources. Contextually relevant material or adaptations to existing content will also be developed with guidance from a community advisory group (CAG).

The intervention arm will thus include the standard of care activities and:

- ► NBO administered at recruitment with feedback and anticipatory guidance.
- ► Rapid behavioural feedback with individualised guidance on infant's movement behaviours and mother–infant interaction styles.
- ► Mental health screening with referrals and individual support as needed for high-risk mothers.
- ► Health literacy content.
  - – Weekly theme-focused content delivered via the app.
  - – Telephonic or in person conversations with discussions around key themes every 2 months and referrals as necessary.

## Intervention delivery

The intervention has been informed by the UK Medical Research Council (MRC) Guidelines for Complex Interventions and is grounded in behaviour change techniques developed by Michie *et al*.[42] We used the Template for Intervention Description and Replication Checklist. Figure 3 details the logic model to illustrate the pathways to impact that guided the intervention approach. The agents of change are community health helpers (HHs), who do not need to have any specific qualification or expertise in health promotion but will be trained by the study team in community health promotion, as well as relevant intervention methodology. The HHs are supported by a supervisor with access to a referral network.

The intervention will be delivered by a combination of phone calls, in person visits, app-based health literacy content, personalised behavioural feedback and guidance, as well as individual counselling sessions for high-risk mothers. The content delivery will be guided by a schedule following key themes according to infant age and developmental stage. Phone calls and in person visits will be guided by a conversation flow chart, which will start with HHs asking mothers how they are doing and gently introducing the theme for discussion using open-ended questions. The flow chart will be designed to allow HHs to either ask follow-up questions, or to provide anticipatory guidance and direct them to relevant resources in the app based on mothers' individual responses (example

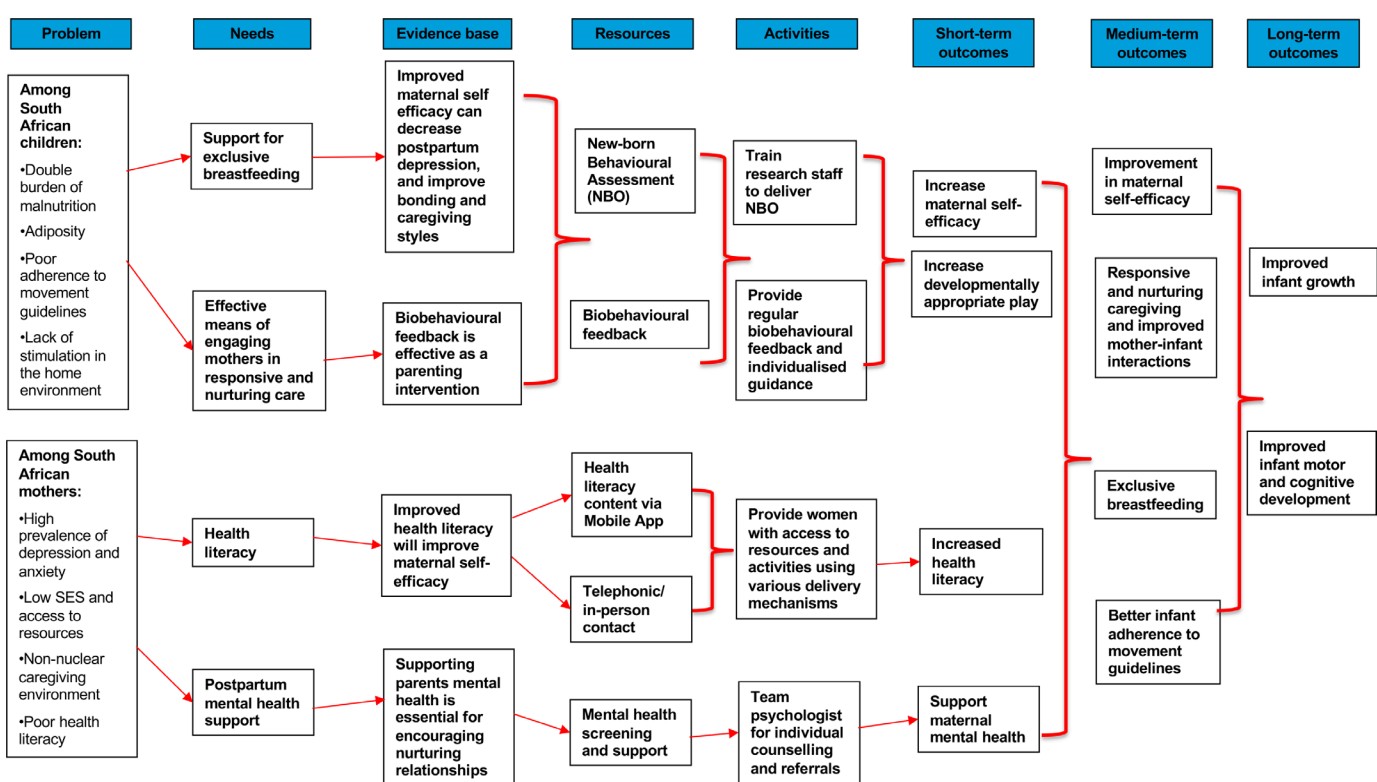

**Figure 3** Play Love And You Study logic model.
**SES - socioeconomic status**

provided in online supplemental figure 2). This will allow mothers to guide the conversation, while still following key themes for discussion. HHs will refer mothers as necessary if any concerns are raised during these conversations.

## Outcome measures
Data collection will be conducted by independent, trained research assistants. The primary outcome measure is efficacy of the intervention in improving maternal self-efficacy at 12 months. Secondary outcomes will estimate efficacy of the intervention in improving infant development at 12 months, and will assess the feasibility and acceptability of the intervention at 6 and 12 months, as well as interim outcomes to determine mechanisms of impact (figure 2).

### Primary outcome at 12 months
► Maternal self-efficacy at 12 months compared between intervention and control groups.

### Secondary outcomes at 6 months
► Breastfeeding self-efficacy from 0 to 6 months compared between intervention and control groups.
► Exclusive breast feeding from 0 to 6 months compared between intervention and control groups.
► Feasibility and acceptability of the breastfeeding intervention.

### Secondary outcomes at 12 months
► Infant development at 12 months compared between the intervention and control groups using the Ages and Stages Questionnaire (ASQ3).
► Feasibility and acceptability of the health literacy and behavioural feedback intervention.
► Change in infant weight to length ratio compared between intervention and control groups.
► Infant fat mass index (FMI) (assessed using dual X-ray absorptiometry (DXA) fat mass divided by length) compared between intervention and control groups.
► Change in maternal health literacy from 6 to 12 months compared between intervention and control groups.
► Change in maternal mental health from 0 to 12 months compared between the intervention and control groups.
► Infant movement behaviours compared between intervention and control groups.
► Mother–infant interaction styles compared between intervention and control groups.

## Data collection
### Primary outcome measure
Maternal self-efficacy will be assessed using the Perceived Maternal Parenting Self-Efficacy Tool at 6 and 12 months. This questionnaire assesses four subscales of self-efficacy: caretaking procedures, evoking behaviour, reading or signalling behaviours and situational beliefs. While this tool was originally designed for assessing mothers'

perceptions of their ability to understand and care for their hospitalised preterm neonates as well as being sensitive to the various levels and tasks in parenting,[52] it has since been successfully used in older, healthy infants in low-income settings.[53] It has also been piloted in a sample of mothers from Soweto, providing a Cronbach's Alpha of 0.95 (data not yet published).

### Secondary outcome measures
Infant development will be assessed at 12 months (and at 6 and 8 months) using the ASQ3. These questionnaires assess development in five domains: communication, gross motor, fine motor, problem-solving and social–personal. The ASQ3 has been successfully applied in community-based, low-resource settings and has been found to be feasible in South Africa.[54 55]

The acceptability of each phase of the intervention will be assessed based on the Theoretical Framework of Acceptability.[56] This framework includes the following criteria: affective attitude, burden, ethicality, intervention coherence, opportunity costs, perceived effectiveness, and self-efficacy. These criteria will be assessed by means of questionnaires specific to each intervention theme, followed by focus group discussions (FGDs) in a subsample of participants to expand on core themes identified.

Feasibility of each phase of the intervention will be assessed by determining whether the developed intervention is implementable in this context, has potential to work, as well as how it could be improved for future use and in other settings. We will follow the MRC guidance on process evaluation of complex interventions.[57] The evaluation will focus on implementation, mechanisms of impact and context.
► Compliance will be assessed by counts (eg, login frequency on app, delivery of messages and monitoring of phone calls using a log) and by discussions with the HHs.
► Intervention dose will be monitored by recording the frequency and duration of contact as well as the topics of conversation between participants and HHs or feedback given to participants using the logs.
► To understand the mechanisms through which the intervention brings about change in specified outcomes, women's experiences of receiving the intervention, contact with HHs and the app will be assessed using questionnaires and focus groups at the close of each intervention phase. We will also use the secondary outcomes (process data) to determine whether individual proposed pathways of impact are effective prior to analysing trial outcomes.
► In considering the context, identification of factors that might act as barriers or facilitators to intervention implementation or effects will be assessed. This will include local and national policy, local service configuration and provision, and sociodemographic and environmental factors of the participants. Our approach to assessing context will be through a combination

of policy mapping, quantitative data collection and longitudinal consultation with the CAG.

Maternal breastfeeding self-efficacy will be assessed every 2 months from 0 to 6 months using the Breastfeeding Self-Efficacy Short Form.[58] This comprises 14 questions, each with a 5-point Likert response scale ranging from 1=not at all confident to 5=very confident. The higher the final aggregate score (which has a range of 14–70), the higher the level of breastfeeding self-efficacy.

Exclusive breast feeding will be assessed every 2 months from 0 to 6 months by asking mothers whether they are currently breast feeding and if not, at what infant age did they stop breast feeding and why they stopped breast feeding. They will also be asked if they are giving their infant anything other than breastmilk, and if so, why.

Weight and length will be measured by trained research staff at baseline and 12 months (and at 4 and 8 months). Weight-to-length ratio (kg/m) will then be calculated to estimate anthropometric changes in body composition. At 12 months, DXA scans will be performed by a trained technician. Whole-body measurements of fat mass and fat-free mass will be extracted for use in analyses. FMI (kg/$m^3$) will be calculated from these estimates to describe adiposity.

Health literacy will be assessed at 6 and 12 months using the parent version of the Health Literacy Questionnaire,[59] which covers nine areas of health literacy including functional health literacy and has been validated in high-income and low-to-middle-income countries.

Various aspects of maternal mental health will be assessed at baseline, 4 and 12 months. The PCL-5[60] will be administered whereby rating scale descriptors include: 'not at all', 'a little bit', 'moderately', 'quite a bit' and 'extremely'. A total symptom severity score is calculated by adding up the item scores, which range from 0 to 80. The cut-off for PTSD that will be used in this study was 31. The EPDS[61] will be used to detect postnatal depression. Scores for each item range from 0 to 3 on a Likert scale and are added together to obtain an overall total. Scores of 7–13 indicate mild depression, 14–18 moderate and 19–30 severe depression. The ACE Questionnaire[62] will assess childhood abuse, neglect and household challenges. A score from 0 to 10 is calculated based on how many types of events were experienced before 18 years old. The ASQ will be administered to assess stressful life events. An additional nine items about social support will be included. The Postpartum Bonding Questionnaire[63] will be used to detect early mother–infant bonding. It consists of four subscales 'impaired mother/infant bonding', 'rejection and anger', 'anxiety about care' and the 'risk of abuse'. These questions are rated on a 6-point Likert scale, ranging from 0 (always) to 5 (never).

Infant adherence to movement guidelines will be assessed every 4 months using accelerometery. Seven-day, 24-hour wrist worn accelerometery data (Axivity-AX3 worn on both mother and infant) will be collected, downloaded (omgui, Open Movement, UK), auto-calibrated and processed. Data is summarised to generate average acceleration (mg) per 5 s interval, as well as overall time (hourly and weekly) spent in varying intensities of activity, which will allow for determination of adherence to movement guidelines and changes in activity levels over the intervention period.

Mother–infant interactions will be measured during play using first-person observation and behavioural analysis every 4 months. Mother–infant interactions have primarily been measured using questionnaires or third-party observation, yet these measures are limited by their subjectivity and intrusiveness. First-person observation using headcams—small cameras which are attached to headbands on both infants and mothers for a period of time while interacting at home—provide a first-person view of the interaction from both infant and mother perspectives with more naturalistic and detailed measures. These headcams have been piloted in South Africa and were found to be largely acceptable and feasible in this context.[64] Headcam measures will be collected during at least five, 3–5 min play interactions over the course of a week. Headcam observations will be coded using Noldus Observer XT software. An event-based coding framework will be used to categorise second-by-second key maternal and infant behaviours, that is, touch, play, facial expressions, hand movement, and gaze. This allows for determination of the probability of certain behaviours preceding or following another, associations between maternal/infant measures and these behaviours, and synchrony between mothers and infants as a potential indicator of sensitivity and responsiveness.

## Patient and public involvement

Prior to trial implementation, a CAG (n=16) of mothers from Soweto with infants aged 0–3 months will be established to serve as experts on the lived experience of being a mother to an infant in Soweto. For about half of these women, family matriarchs will be recruited to form a separate CAG (n=8). Research from South Africa has indicated that family matriarchs play an important role in the choices that mothers make around infant feeding and nutrition,[13] and as such it is pertinent they be included in interventions targeting breast feeding and other child care practices.

Intervention content and activities, and behavioural feedback will be tested and/or developed with the CAG, using FGDs and piloting of intervention content and feedback. Thus, the initial acceptability and feasibility of the intervention content will be tested by the CAG in order to understand any changes required, and thus refine the intervention accordingly.

The CAG will continue to advise throughout the duration of the intervention, via regular (every 4 months) FGDs where the research team will raise queries about intervention content, delivery and uptake, retention strategies, and feedback mechanisms. Additionally, the CAG will provide important insight into their lived experience of being a mother in Soweto over time and as their infant grows. This will provide useful insight to guide

intervention conversations and to better understand the context within which the intervention is being delivered.

## Monitoring

Monitoring of intervention activities will be done throughout the trial. The delivery of each contact (duration, date, content) will be recorded using a monitoring log in order to be able to monitor intervention intensity and frequency. Engagement with the app content and behavioural feedback (ie, number of page visits per day and duration of time spent on the app) will be monitored via the app.

Adherence will be assessed every 4 months, and strategies to improve adherence will be discussed with the CAG as necessary. Additionally, the research team will meet with HHs weekly to determine any concerns or issues with compliance or retention, and strategies will be discussed accordingly. Any adverse events will be recorded and reported as necessary according to standard operating procedures.

## Sample size

Given expected attrition of 30%, recruiting 210 mother–infant pairs should yield a final sample size of 150 mother–infant pairs, which will be divided into 2 groups (n=75 each). A sample size of 150 mother–infant pairs divided into 2 groups (n=75 each; control and trial group) will allow for 99% power to detect efficacy of the intervention based on maternal self-efficacy data assessed in the piloting phase of the behavioural feedback intervention (data not yet published).

Attrition will be minimised by maintaining regular contact with both groups, by recruiting participants who plan on remaining in Soweto, and by using a flexible and versatile delivery approach including in-person visits, app-based content, telephonic sessions, mobile messaging and site visits.

## Data analysis plan

The efficacy of the intervention will be assessed using intention to treat analysis. Secondary outcome data will be compared between the intervention and standard of care arms using Student's unpaired t-tests. Focus will be placed on the primary efficacy outcome (maternal self-efficacy) due to limited power for secondary outcomes, which will therefore be treated with caution and presented using point estimates and precision (95% CI). Sensitivity analyses will include comparing baseline characteristics between intervention and standard of care arms, examining the effect of intervention dose on outcome data and stratifying analyses by sex.

The feasibility and acceptability of the intervention at each phase will be assessed using qualitative and quantitative methods. FGDs will be transcribed and coded line by line, and data will be grouped according to common themes. Themes will be sorted into subthemes and continually reviewed to identify an overarching narrative to identify key findings from each FGD.

## Data management

REDCap will be used for data management (https://www.project-redcap.org). Secure Sockets Layer (SSL) provides the needed security for the REDCap similar to what a Virtual Private Network (VPN) will provide. The Wits University server hosting REDCap is resilient and hardened against hacking and other forms of attacks. All data traffic is encrypted, and an installed network firewall provides secure network access. The REDCap system provides inherent application-level security including access controls and comprehensive audit trails, and all incoming requests get intentionally filtered, sanitised and escaped to prevent against attacks.

Data files and metadata will be uploaded and stored on REDCap by the project data manager, on a password-protected Wits server for the duration of the project to enable team access. REDCap is used for data and metadata storage, as well as for data collection. Data will be inputted online with secure web authentication, data logging and SSL encryption. Accelerometery files and video files, as well as FGD notes and transcripts, will be stored on a secure Wits server, as well as on an external hard drive, and backed up as metadata to REDCap. Regular backup copies will be made by the project data manager on a password-protected file space. When the project is completed, the full database will be stored on both Developmental Pathways for Health Research Unit (DPHRU) and University servers. The items and metadata on the institutional repository are backed up on a daily basis and these backups are kept on tape. All data collection devices will be encrypted.

Data from this study will be stored for a minimum of 10 years from the completion of the study in order to allow for any verification of data. Long-term data storage will be in .csv files using a standard (simple) American Standard Code for Information Interchange (ASCII) coding scheme in line with UK Data Service recommendations (https://www.ukdataservice.ac.uk/manage-data/format/recommended-formats).

## Ethics and dissemination

Ethical approval has been granted for the establishment of the CAG (M210846), and for the trial (M220217) from the Human Ethics Research Committee of the University of the Witwatersrand, Johannesburg, South Africa. Additional approval has been granted by the Research Committee of Johannesburg Health District (GP_202202_021). Participants will be provided with an information sheet and required to provide written consent (online supplemental document 1) prior to being enrolled. Study results will be shared via publication in peer-reviewed journals, conference presentation and media engagement.

## DISCUSSION

If found to be feasible, acceptable and successful, these findings could lead to implementation of the intervention

on a larger scale. Particularly, implementing intervention content within primary healthcare services, and linkage with existing digital solutions (such as MomConnect in South Africa), would allow for dissemination of the health literacy components of this intervention nationally. Furthermore, the PLAY Study app has the potential to be developed further and made into an accessible app for all mothers. All of this content aligns with the Nurturing Care Framework and WHO guidelines and is therefore relevant at a national policy level, as well as internationally. Additionally, engagement with the community and dissemination of findings to participants and other stakeholders provides opportunities for knowledge mobilisation and therefore greater impact. There is potential to transfer the intervention to other LMICs and high-income countries; however, since this content was designed with the Soweto community to be contextually relevant, adjustments would need to be made to ensure the feasibility and acceptability in other contexts. Certain aspects of the intervention, such as the accelerometery and headcam assessments and feedback, are more complex and would require funding and training to implement on a larger scale, yet the concept of providing feedback using some of the other assessments such as development and growth can feasibly be incorporated into digital platforms or the PLAY Study app for scaling up.

The PLAY Study relies to some extent on the availability of cell phone or computer access for the delivery of intervention content, which may be limiting in the Soweto setting. However, we will attempt to mitigate this by making the app data free, by making the app available on multiple platforms (web, mobile, android, Apple etc) and by continuously consulting with the CAG to ensure we come up with appropriate solutions if we encounter lack of access. Findings from this study are relevant to a socioeconomically disadvantaged group of mothers, and thus cannot be extrapolated to higher income settings or different population groups. This study does not specifically include fathers due to the low prevalence of caregiving fathers in this context. Future studies directed at both parents or at other caregivers would need to be adjusted for those population groups.

**Author affiliations**
[1]SAMRC/Wits Developmental Pathways for Health Research Unit, University of the Witwatersrand, Johannesburg, South Africa
[2]Centre for Academic Mental Health, Addiction and Suicide Research, School of Social & Community Medicine, University of Bristol, Bristol, UK
[3]DSI-NRF Centre of Excellence in Human Development, University of the Witwatersrand, Johannesburg, South Africa
[4]Department of Social Medicine, MRC Integrative Epidemiology Unit, University of Bristol, Bristol, UK

**Acknowledgements** The authors would like to thank the Wellcome Trust and the National Institute for Health Research for their financial support which made this study possible.

**Contributors** AP is the principal investigator for the PLAY Study. AP, SAN, RP, LR and DL conceptualised the study design. AP wrote the initial draft of the protocol. SAN, RP, LR, DL, DM, FB, SJC and HT read and contributed to the final version. All authors provided edits and critiqued the manuscript for intellectual content.

**Funding** This work was supported by the National Institute for Health Research (NIHR) (using the UK's Official Development Assistance Funding) and Wellcome (222007/Z/20/Z) under the NIHR-Wellcome Partnership for Global Health Research. The views expressed are those of the authors and not necessarily those of Wellcome, the NIHR or the Department of Health and Social Care.

**Competing interests** None declared.

**Patient and public involvement** Patients and/or the public were involved in the design, or conduct, or reporting, or dissemination plans of this research. Refer to the Methods and analysis section for further details.

**Patient consent for publication** Not applicable.

**Provenance and peer review** Not commissioned; externally peer reviewed.

**ORCID iDs**
Alessandra Prioreschi http://orcid.org/0000-0002-6913-0706
Shane A Norris http://orcid.org/0000-0001-7124-3788

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
