## [Reviewer comments · BMJ Open]

ARTICLE DETAILS

TITLE (PROVISIONAL)	Protocol for the PLAY study: a randomised controlled trial of an intervention to improve infant development by encouraging maternal self-efficacy using behavioural feedback
AUTHORS	Prioreschi, Alessandra; Pearson, Rebecca; Richter, Linda; Bennin, Fiona; Theunissen, Helene; Cantrell, Sarah J; Maduna, Dumsile; Lawlor, Deborah; Norris, Shane

VERSION 1 – REVIEW

REVIEWER	Provenzi, Livio Scientific Institute IRCCS E. Medea, 0-3 Center for the at-Risk Infant
REVIEW RETURNED	28-Aug-2022

GENERAL COMMENTS	1. Is sleep a movement category? Is sedentary behavior just the opposite of physical activity? I have doubts on the category of movement reported by authors at page 4. Is there a specific rationale for this? If yes, please report it.2. Biobehavioral feedback is not properly framed and introduced in the Introduction of the manuscript. It is just mentioned at the end of the Introduction - please, frame it better to help the readers understand what you mean with this.3. Hypotheses and aims are generally reported. It would be better to have specific aims and relative hypotheses reported one by one, according to the study design and outcome measures.4. The intervention and the paper focus on the mother - this reflects a general cultural tendency to consider mothers - and in many cases, they really are - as the primary caregivers. Can the intervention be delivered to primary caregivers in general? To fathers? I suggest the authors to use caregiver instead of mother if they can; or to discuss how the intervention can be applied in general not only with mothers rather with primary child caregivers.4. I would anticipate when sample size is firstly mentioned that the sample size estimation has been reported later in the manuscript.5. Some of the thematic core elements of the intervention arm are not introduced in the background of the article: for instance, there is no discussion of the importance of exclusive breastfeeding for infants' movement and healthy development.6. The authors refer to personalized guidance - nonetheless, it is not clear how the personalization will occur.7. It is not clear what the "bio" refers to in "Biobehavioral" feedback. It seems that behavioral feedback might be appropriate here.8. It is mentioned that head-mounted cameras will be used every three months - notwithstanding it is not clear for how much time, when, how they will be used, how data will be collected. In
--

	general, the data collection structure behind the study is poorly described. 9. There is also little description of the App and how many times the experimenters/clinicians will interact with parents with app and/or phone. 10. What do the authors mean when they say "Each contact will be recorded" - audiotaped? videotaped? Or just taking note of timing and frequency of meetings? 11. It is not clear which variables will be collected to provide a feedback to parents. Sleep is mentioned in the introduction and in supplementary figure 1, but there is mention of sleep variables collection in the methodology. 12. Please, check your figures for background. Some of them (Fig 1 and Fig 2) probably have a blank background and they appear "black" 13. Figure 3 - what does the vertical axis mean? The boxes are distributed randomly on the vertical axis or is there a rationale? They seem to be misaligned. 14. Limitations and/or Risk-and-mitigation section is missing in the main text.
--	--

REVIEWER	Kvestad, Ingrid NORCE Norwegian Research Centre AS
REVIEW RETURNED	10-Sep-2022

GENERAL COMMENTS	I would like to congratulate the authors on a very nice and interesting study, as well as the clear and interesting description in this protocol paper. It would be interesting if the authors added a paragraph in the far end on the potential implications of the findings. Could the intervention be implemented on a larger scale? Locally – in other LMIC settings? Is it too complex or would this be feasible. Culturally acceptance – require adjustments for other settings, or is the intervention more universal? In this section, perhaps also address limitation of the study? Is blinding possible? I understand that you aim for the measurement team to be blinded, but strictly speaking this is not a blinded study. Perhaps describes measures to ensure as far as possible that the testers will not know which intervention group the infant was in? I think you should specify that this is not a blinded study however. On page 5-6, Study design, it is a little confusing that you state 75 in each group and then that you will recruit 210 pairs. I understand from the text below that this is to ensure 75 pairs, but please specify also in this section, Page 9: some references to the mental health measures? Page 11: Primary outcome: Please specify what is the primary outcome – is there a of the assessment package? Or will one of the subscores be the primary outcome? This should be explicit in the text. Good luck with the study!
---

VERSION 1 – AUTHOR RESPONSE

Reviewer: 1

Dr. Livio Provenzi, Scientific Institute IRCCS E. Medea

Comments to the Author:

1. Is sleep a movement category? Is sedentary behavior just the opposite of physical activity? I have doubts on the category of movement reported by authors at page 4. Is there a specific rationale for this? If yes, please report it.

Yes, sleep is one of the three categories of movement behaviours (along with physical activity and sedentary behaviour), which are well recognised in movement research. The basis of this is that these three behaviours make up all of the movement a person/child can do in a 24 hour day, and that if one behaviour is increased, a decrease must happen in one of the other two behaviours thus they are contingent on each other. Guidelines for these three movement behaviours exist globally (in South Africa, Canada, Australia) and have been reiterated by the WHO (all references below).

Similarly, sedentary behaviour is considered a separate category of behaviour, and is not just the opposite of physical activity or the same as physical inactivity. Sedentary behaviour is defined as any behaviour in which the corresponding energy expenditure is ≤ 1.5 metabolic equivalents (metabolic equivalent of task [MET]) in a sitting, reclining, or lying position while at rest. Physical inactivity is defined as insufficient levels of the practice of physical activity, as recommended by the WHO for each age range. Thus, physical activity or inactivity (insufficient physical activity) is one behaviour which has unique guidelines and health benefits, while sedentary behaviour has a separate set of guidelines and health implications. These behaviours are independent of each other, simply adhering to physical activity guidelines does not protect one from the health implications of spending too much time sedentary, and an individual can be physically active enough, yet still be too sedentary. Literature on sedentary behaviour as its own category of movement is also listed below.

We have added slightly more detail on this in the background, but have provided further reference to the literature as it is already well described.

Movement guidelines:

- <https://www.who.int/publications/i/item/9789241550536>
- *Okely, A.D., et al. A collaborative approach to adopting/adapting guidelines. The Australian 24-hour movement guidelines for children (5-12 years) and young people (13-17 years): An integration of physical activity, sedentary behaviour, and sleep. <https://doi.org/10.1186/s12966-021-01236-2>*
- *Tremblay MS., et al. Canadian 24-Hour Movement Guidelines for Children and Youth: An Integration of Physical Activity, Sedentary Behaviour, and Sleep. <https://doi.org/10.1139/apnm-2016-0151>*
- *Draper CE., et al. The South African 24-Hour Movement Guidelines for Birth to 5 Years: An Integration of Physical Activity, Sitting Behavior, Screen Time, and Sleep. <https://doi.org/10.1123/jpah.2019-0187>*

Sedentary behaviour:

- <https://www.who.int/publications/i/item/9789240015128>
- *Sedentary Behaviour Research network. Letter to the Editor: Standardized use of the terms “sedentary” and “sedentary behaviours”. <https://doi.org/10.1139/h2012-024>*
- *Tremblay MS, Aubert S, Barnes JD, et al. Sedentary Behavior Research Network (SBRN) - Terminology Consensus Project process and outcome. doi:10.1186/s12966-017-0525-8*

2. Biobehavioral feedback is not properly framed and introduced in the Introduction of the manuscript. It is just mentioned at the end of the Introduction - please, frame it better to help the readers understand what you mean with this.

More information on feedback as a behaviour change technique in general has now been included before describing feedback in parenting interventions specifically.

3. Hypotheses and aims are generally reported. It would be better to have specific aims and relative hypotheses reported one by one, according to the study design and outcome measures.

We have rewritten the aims to clarify the specific aims for each phase of the intervention, and have rewritten the corresponding specific hypotheses.

4. The intervention and the paper focus on the mother - this reflects a general cultural tendency to consider mothers - and in many cases, they really are - as the primary caregivers. Can the intervention be delivered to primary caregivers in general? To fathers? I suggest the authors to use caregiver instead of mother if they can; or to discuss how the intervention can be applied in general not only with mothers rather with primary child caregivers.

While we agree that there is a general tendency to focus on mothers and neglect the impact that fathers have on caregiving practices, in the South African context including fathers is logistically problematic since very few children are raised with a father present in the home. For example in South Africa only 38% of children have a father residing in the home with them (and in the poorer regions, such as Soweto where our study is being conducted, only 17% live with a father), while over 90% of children under two live with their mother. Thus in our context the primary caregiver is usually the mother. Alternatively, when the mother is not the primary caregiver children are raised by grandparents (68%), aunts/other relatives (19%), or siblings (7%), or other (6%) – this makes it logistically very difficult to tailor an intervention to any primary caregiver when that caregiver may be an adult, an elderly person or a child. Thus in order to test the feasibility, acceptability and efficacy of such an intervention, we need to start with targeting the usual primary caregiver (which in 90% of cases is the mother), whereafter future adaptations can attempt to target other potential caregivers, but that would need trial development and piloting and establishment of different groups of community advisors. We have added this as a limitation.

To try and understand the impact of the second largest group of caregivers (grandparents), we have established a community advisory group of matriarchs to the mothers (described in the protocol), and a sub study not described in this protocol will run a mini intervention targeting those matriarchs as well to see if doing so changes the feasibility, acceptability and efficacy of the trial.

Reference: Hall K, Richter L, Mokomane Z & Lake L (eds) (2018) South African Child Gauge 2018. Cape Town: Children's Institute, University of Cape Town

5. I would anticipate when sample size is firstly mentioned that the sample size estimation has been reported later in the manuscript.

The sample size estimation appears on page 13 of the manuscript under the heading "Sample size"

6. Some of the thematic core elements of the intervention arm are not introduced in the background of the article: for instance, there is no discussion of the importance of exclusive breastfeeding for infants' movement and healthy development.

Thank you for this valuable comment, we have expanded greatly on the introduction to ensure we properly introduce and contextualise each intervention element. We have also restructured the introduction to help with ease of reading in line with the intervention themes.

7. The authors refer to personalized guidance - nonetheless, it is not clear how the personalization will occur.

We have now detailed how the feedback will be presented and how it will be personalised to each individual's own data. This is also discussed in response to another comment further down.

8. It is not clear what the "bio" refers to in "Biobehavioral" feedback. It seems that behavioral feedback might be appropriate here.

We have reworded to "behavioural feedback" throughout

9. It is mentioned that head-mounted cameras will be used every three months - notwithstanding it is not clear for how much time, when, how they will be used, how data will be collected. In general, the data collection structure behind the study is poorly described.

Thank you for this comment, we agree and have now added a data collection section to describe the data collection procedures for each primary and secondary outcome measure. We hope this helps with understanding the measurements.

10. There is also little description of the App and how many times the experimenters/clinicians will interact with parents with app and/or phone.

We have now added more detail about the App, and the frequency of engagement with the App under the intervention section. We have additionally reworked the intervention components to add more detail and make them clearer.

11. What do the authors mean when they say "Each contact will be recorded" - audiotaped? videotaped? Or just taking note of timing and frequency of meetings?

We have clarified that each contact between intervention delivery team and participants will be recorded and detailed in a monitoring log

12. It is not clear which variables will be collected to provide a feedback to parents. Sleep is mentioned in the introduction and in supplementary figure 1, but there is mention of sleep variables collection in the methodology.

Thank you for pointing this out. We have now added more detail on the type of data and information that will be collected and presented back to parents, and have added examples of what the graphical feedback would look like. We have also discussed how the guidance will be personalised to their own data.

13. Please, check your figures for background. Some of them (Fig 1 and Fig 2) probably have a blank background and they appear "black"

Thank you, we have redone and replaced all of the figures

14. Figure 3 - what does the vertical axis mean? The boxes are distributed randomly on the vertical axis or is there a rationale? They seem to be misaligned.

We have fixed this figure to make it more coherent.

15. Limitations and/or Risk-and-mitigation section is missing in the main text.

A limitations section has also been added (and there is a Strengths and Limitations section as per journal guidelines at the beginning of the manuscript).

Reviewer: 2

Dr. Ingrid Kvestad, NORCE Norwegian Research Centre AS

Comments to the Author:

I would like to congratulate the authors on a very nice and interesting study, as well as the clear and interesting description in this protocol paper.

1. It would be interesting if the authors added a paragraph in the far end on the potential implications of the findings. Could the intervention be implemented on a larger scale? Locally – in other LMIC settings? Is it too complex or would this be feasible. Culturally acceptance – require adjustments for other settings, or is the intervention more universal?

In this section, perhaps also address limitation of the study?

Thank you for the suggestion. We have added a section which discusses the potential for scale-up of this study, and how our engagement with the community and dissemination of results can assist with knowledge mobilisation. A limitations section has also been added (and there is a Strengths and Limitations section as per journal guidelines at the beginning of the manuscript)

2. Is blinding possible? I understand that you aim for the measurement team to be blinded, but strictly speaking this is not a blinded study. Perhaps describes measures to ensure as far as possible that the testers will not know which intervention group the infant was in? I think you should specify that this is not a blinded study however.

We have added detail to the study design section to explain that the data collection team for baseline, and outcome assessments will be blinded, since they will work independently of the intervention teams. One team will collect data before participants have been randomised (at baseline) and a second blinded team will collect outcome data. Randomisation into either group will only happen after baseline assessments are completed. However the intervention delivery team will not be blinded.

3. On page 5-6, Study design, it is a little confusing that you state 75 in each group and then that you will recruit 210 pairs. I understand from the text below that this is to ensure 75 pairs, but please specify also in this section,

Thank you we have clarified this in the study design section.

4. Page 9: some references to the mental health measures?

These have now been added under the expanded data collection section

5. Page 11: Primary outcome: Please specify what is the primary outcome – is there a of the assessment package? Or will one of the subscores be the primary outcome? This should be explicit in the text.

We have now reworded the entire outcomes section of the paper, and have made the data collection processes for these measures clearer. We have also adjusted all figures to clearly outline the timing of each intervention theme, and the outcome assessments that are planned at different stages, as well as the contact points with participants. We hope this is now clearer and that your comment is addressed within these changes.

6. Good luck with the study!

Thank you very much, and for your helpful comments!

VERSION 2 – REVIEW

REVIEWER	Provenzi, Livio Scientific Institute IRCCS E. Medea, 0-3 Center for the at-Risk Infant
REVIEW RETURNED	21-Nov-2022
GENERAL COMMENTS	Dear authors, thanks for being responsive to my requests and comments. Thanks for providing appropriate information that not only improved the quality of the manuscript (as I can see) but also

	let me learn something new. This is an ambitious study with big potentials, good luck!
--	--

VERSION 2 – AUTHOR RESPONSE

** **

Reviewer: 1

Dr. Livio Provenzi, Scientific Institute IRCCS E. Medea

Comments to the Author:

Dear authors, thanks for being responsive to my requests and comments. Thanks for providing appropriate information that not only improved the quality of the manuscript (as I can see) but also let me learn something new. This is an ambitious study with big potentials, good luck!

Thank you for the positive feedback and for your helpful reviews!

Additional changes

In addition to the requested changes we have formatted some minor errors, and have changed the primary outcome to maternal self-efficacy. This is based on continued piloting with our community advisory group while awaiting decision on this manuscript, which has convinced us that the mechanism through which we would be able to ultimately impact infant development with this intervention is via improving maternal self-efficacy. Therefore the trial will now primarily assess efficacy in improving maternal self-efficacy, and secondarily estimate efficacy in improving infant development. We have also changed the infant development measurement tool from the Bayley Scales to the ASQ3. This is because upon further investigation we have found that it will not be feasible to train our community health helpers in conducting the Bayley Scales to the required standards. The ASQ3 has been successfully applied in community-based, low-resource settings, and has been found to be feasible in South Africa. It is easy to administer by providers of varying levels of education and expertise, and requires minimal training.